# Development of a digital imaging analysis system to evaluate the treatment response in superficial infantile hemangiomas

**Mingfeng Xie**[1,2,3☯], **Jianping Liu**[1☯], **Pingsheng Zhou**[2,3], **Xianyun Xu**[2,4], **Haijin Liu**[1], **Linshan Zeng**[1], **Feng Chen**[1], **Yong Zeng**[1], **Haijin Huang**[1], **Wei Peng**[1], **Hui Xiao**[2,3], **Qian Liu**[2,3]*

1 Jiangxi Provincial Clinical Research Center for Vascular Anomalies, The First Affiliated Hospital of GanNan Medical University, Ganzhou, Jiangxi, China, 2 Chinese & Western Integrative Medicine Discipline, Jiangxi University of Chinese Medicine, Nanchang, Jiangxi, China, 3 Jiangxi Key Laboratory of TCM for Prevention and Treatment on Hemangioma, Nanchang, Jiangxi, China, 4 Jiangxi University of Chinese Medicine, Nanchang, Jiangxi, China

☯ These authors contributed equally to this work.
* liuqianjxzyy@163.com

**Data Availability Statement:** All relevant data are within the paper and its Supporting Information files.

## Abstract

Superficial infantile hemangiomas (IH) are benign vascular tumors common in children characterized by bright red "strawberry" lesions on the skin. In order to optimize the treatment for this disease, there is a need to develop objective tools to assess treatment response. Since a color change in the lesion is a good indicator of treatment response, we have developed a digital imaging system to quantify the values of red, green, and blue (RGB) difference and RGB ratio between the tumor and normal tissue to take into account the variations in color between different skin types. The efficacy of the proposed system in assessing treatment response in superficial IH was evaluated in relation to established visual and biochemical tools used to grade hemangiomas. As the treatment progressed, the RGB ratio was almost 1, while the RGB difference was close to 0, which indicates a good response to treatment. There was a strong correlation between the RGB score and the other visual grading systems. However, the correlation between the RGB scoring system and the biochemical method was weak. These findings suggest that the system can be used clinically to objectively and accurately evaluate disease progression and treatment response in patients diagnosed with superficial IH.

## Introduction

Hemangiomas are caused by abnormalities in the development of embryonic vascular tissue and regulation of blood vessel formation [1]. Infantile hemangioma (IH) is the most common benign tumor in infants and children [2]. The incidence is even higher in infants who weigh less than 1000g [3, 4]. Infantile hemangioma (IH) can affect any part of the body but occur more frequently on the face and limbs. The appearance of skin hemangioma lesions may range from small red macules to large dome-topped or polypoid papules [5]. Although these

**Funding:** The authors received no specific funding for this work.

**Competing interests:** The authors have declared that no competing interests exist.

haemangiomas are not life-threatening, they can significantly alter the patient's physical appearance and negatively impact the child's quality of life and self-esteem [6].

The hemangiomas in the proliferation stage are divided into three types; superficial, deep type, and mixed [6], according to their degree of invasion. The superficial IH is often called "strawberry hemangioma" due to its typical shape, bright red color, and irregular protrusions on the skin surface [1]. These lesions are usually treated with oral propranolol treatment. Following treatment, the diameter of the tumor, the degree of protrusion, and the blood cavity generally start to decrease gradually. Furthermore, the local skin temperature and tumor color also change [4, 7]. Several visual grading systems have been proposed to assess the treatment efficacy for IH, including the Achauer method, the visual analog scoring (VAS) method, and the infant hemangioma activity scoring method (HAS) [8–11]. Ultrasound, computed tomography (CT), and magnetic resonance imaging (MRI) can also be used to assess the extent of the disease [12–14]. Furthermore, quantitative methods that measure the skin temperature [14, 15] or the body fluid biochemical index have also been used [15, 16].

A change in the color of the tumor is also an important indicator of treatment response in IH [8]. However, there is currently no accurate, effective, and objective method to assess changes in color. As a result, the clinicians have to rely on a visual evaluation [17]. In order to overcome this problem, we aimed to develop a digital imaging system to objectively measure and analyze changes in the color of IH following treatment. The efficacy of the system in grading the severity of IH was compared with established visual and quantitative grading tools.

## Materials and methods

### Design of the digital imaging system

The digital imaging system developed in this study consisted of three parts; a light source system (light source box, hood), an imaging system (CDC camera), and a software processing system (computer) [18–20]. The system was assembled as shown in Fig 1. A standard D65 light source was used to acquire the images on the CDC camera [21]. After comprehensive consideration, we adopt Olympus, model D33235 digital camera for the imaging system, with a total pixel of 10 million and an effective pixel of 9.3 million. We use M.Ziuiko professional lens, and the sensor is a 17.4mm*13.0mm CCD. For the light source system, we chose P120 light source

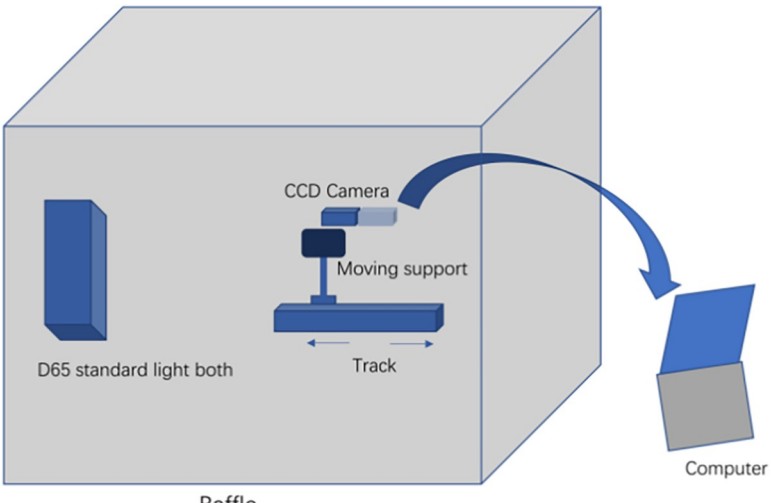

**Fig 1. Digital imaging analysis system.**

box jointly produced by TILO and 3nh Company and PHILIPS TLD-36W/865 three-color fluorescent tube produced by PHILIPS Company as light source. The model of this light source is G13, which can produce light with a light flux of 3350, color rendering index of 85, color temperature of 6500K, and the required power is 36W.

### Image acquisition

A light source box equipped with two sets of D65 light strip sources was built and covered with white reflectors to ensure that the lesion was uniformly covered with white light. The entire shooting system was placed in a light hood to block the external light and hence minimize the influence of the ambient light on the red, green and blue (RGB) measurements. The researcher adjusted the light hood first and then the camera aperture to ensure enough light reached the detector. Images of the tumor and surrounding normal tissue were then acquired and analysed

### Calibration of RGB measurements

The camera was calibrated to acquire RGB measurements as follows. Images of the tumor and surrounding normal tissue were first acquired, and the RGB parameters were adjusted to ensure that at least one component within the image had an RGB value of 255. Subsequently, a gray standard board was placed over the lesion to adjust the RGB output parameters of the camera to 122, 122, 121. After that, a black standard plate was placed over the lesion (or used to cover the lens). The RGB parameters were all set to 0. Finally, a white standard plate was placed over the lesion, and the camera parameters were adjusted to set all the output RGB values to 255.

### Image analysis using an in-house developed software

In-house developed software was used to grade the superficial IH based on an RGB score. The software quantified the actual tumor color by substracting the RGB intensity of the tumor from the RGB intensity of the first 50 pixels of normal tissue surrounding the lesion. This method was used to account for individual variations in skin color.

### Clinical evaluation of the digital imaging system

The accuracy of the digital imaging system in evaluating the severity of IH was clinically assessed on patients treated for superficial IH as follows.

### Data acquisition

All patients diagnosed with superficial IH and treated with oral propranolol from July 2012 to June 2019 at the First Affiliated Hospital of Gannan Medical University were eligible for the study.

### Diagnostic, inclusion, and exclusion criteria

**Diagnostic criteria.** Infantile haemangioma tend to appears a first few weeks after birth. It generally initially appears as red spots, similar to mosquito bites. The mass gradually enlarges and can protrude on the surface of the skin. The lesion typically has a bright colour which does not fade when pressed. However auxiliary imaging examinations such as ultrasound are required to confirm the diagnosis.

**Inclusion criteria:**

1. According to the VAs classification guidelines issued by ISSVA in 2010–2018, a diagnosis of superficial IH with typical clinical manifestations combined with physical examination and imaging examination;

2. Superficial IH who has not received any form of treatment before admission, with high treatment compliance.

   **Exclusion criteria:**

1. Other vascular abnormalities such as vascular malformations, congenital hemangioma, etc.;

2. The child's IH color Doppler ultrasound suggested deep hemangioma;

3. Cases with poor compliance or withdrawal;

4. The child's condition information is missing;

5. Too few follow-up times (<3 times);

   Any one of the above conditions will be excluded.

## Treatment provided

All patients were treated with oral propranolol. The first treatment was administered in the hospital using a daily dose of 0.5mg/kg. On admission, the clinicians monitored the patient's blood pressure, heart rate, and blood glucose levels to ensure that the patient had no adverse reactions to the drug. Additionally, the color, volume, and texture of the tumor were also recorded. If there were no adverse reactions to the drug, the rest of the treatment was delivered at home. The dose was increased gradually by 0.5mg/kg daily up to a maximum of 2.0 mg/kg daily. The daily dose was divided into three treatments. The child's carers were instructed to provide the medication 30 minutes after meals to enhance the child's tolerance to the medicine and reduce the risk of developing hypoglycemia. Throughout treatment, the child's carers were also instructed to monitor the child's vital signs, observe the child's sleep quality, and pay attention to any diarrhea. If, during treatment, the child developed any side effects, the daily treatment dose was reduced. However, the treatment was stopped immediately in patients who had severe reactions. These patients were admitted to the hospital for treatment and observation for nine days. If there were no obvious adverse reactions to the drug after nine days, the child was discharged and the treatment continued at home. The treatment was stopped until the desired therapeutic effect was obtained.

## Follow-up procedure

The patients were followed up every four weeks until the end of treatment. During each follow-up appointment, the clinicians examined the blood pressure, glucose levels, and heart rate and adjusted the propranolol dose according to the child's weight.

The researcher also acquired images of the tumors using the CDC camera (Fig 2) and analyzed the images using the digital system. The clinicians also graded the severity of the disease using other established methods used to grade IH, including the Achauer et al. [8] method, the HAS system [10], the VAS system [9], and the biochemical blood indicator method.

## Evaluation of treatment efficacy using the digital imaging system

The workflow used to evaluate the treatment efficacy of the digital imaging system is summarized in Fig 3. The comprehensive RGB for a particular region of interest was defined as the average RGB within that area. The RGB average is calculated by dividing the sum of the brightness of each pixel of the image by the total number of pixels. The difference between the comprehensive RGB of the tumor and surrounding normal tissue was calculated before treatment and at monthly intervals after starting treatment til the end of the treatment. The RGB ratio

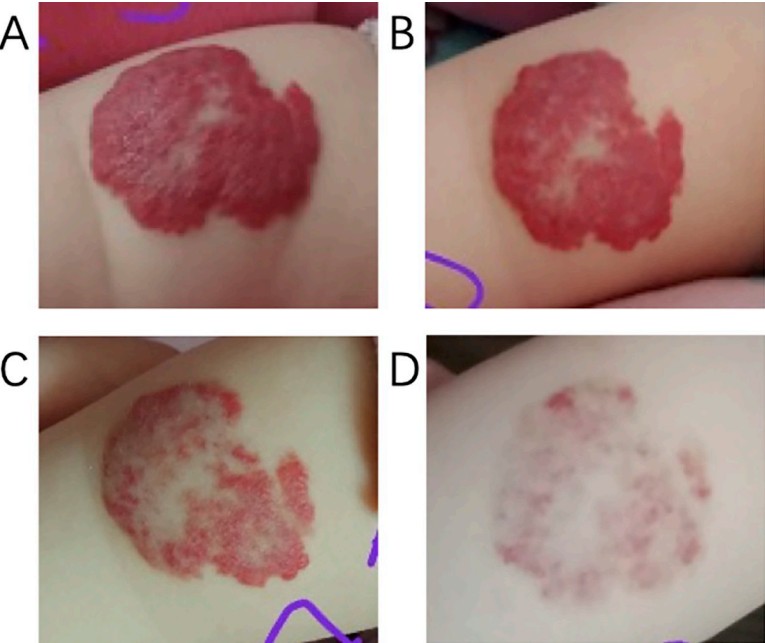

**Fig 2. Photographs of a hemangioma on the left arm.** (A) The photograph acquired before treatment; (B) The photograph acquired one month after treatment; (C) The photograph taken two months after treatment; (D) The photograph acquired at the end of treatment.

was also calculated by dividing the comprehensive RGB value of the tissue around the tumor by the comprehensive tumor RGB. An RGB difference between the tumor and normal tissue of 0 and an RGB coefficient of 1 indicate a perfect treatment response whereby there is no longer any difference in the comprehensive RGB between the tumor bed and the surrounding normal tissue. The data conclude the information of the patient, the hemangioma images and the RGB results will be uploaded and stored.

## Ethical consideration

Approval to conduct this study was sought and obtained from the research ethics committee of the First Affiliated Hospital of Gannan Medical University (Ethics approval no: LLSC-2021062503). Written informed consent was obtained from all the patients' guardians participating in this study.

## Statistical analysis

Statistical analysis was performed using the Statistical Package for the Social Sciences (SPSS) software version 22.0. The clinical and demographic data were expressed as absolute frequencies or means +/- standard deviation (SD). The RGB ratio at different time points was compared using the one-way analysis of variance (ANOVA) test. The Mann-Whitney test was used to identify any statistically significant differences between the RGB score obtained from the digital system and the other grading methods. The correlation between the RGB score and the other established grading method was evaluated using the Pearson test for the continuous data and the Spearman test for the ordinal data. A p-value below 0.05 was deemed statistically significant.

## Results

### Clinical and histological features

A total of 207 patients with superficial IH were included in the study, of which 68 were males and 139 were females. The average age of the patients at their first visit to pediatric surgery was 3.48 ± 1.96 months. The average treatment time was 3.52 ± 1.82 months. Out of the 207 tumors in the study, 61 cases (29.4%) were in the head and neck, 63 cases (30.4%) were in the limbs, 75 cases (36.2%) were in the trunk, and 8 cases (4.0%) were in the perineum. The clinical and demographic characteristics of the patients are summarized in Table 1.

### Comprehensive RGB value before and after treatment

Before treatment, the mean comprehensive RGB values were 90.14±29.35 for the tumor and 146.93±39.57 for the normal tissue around the tumor. After treatment, the mean

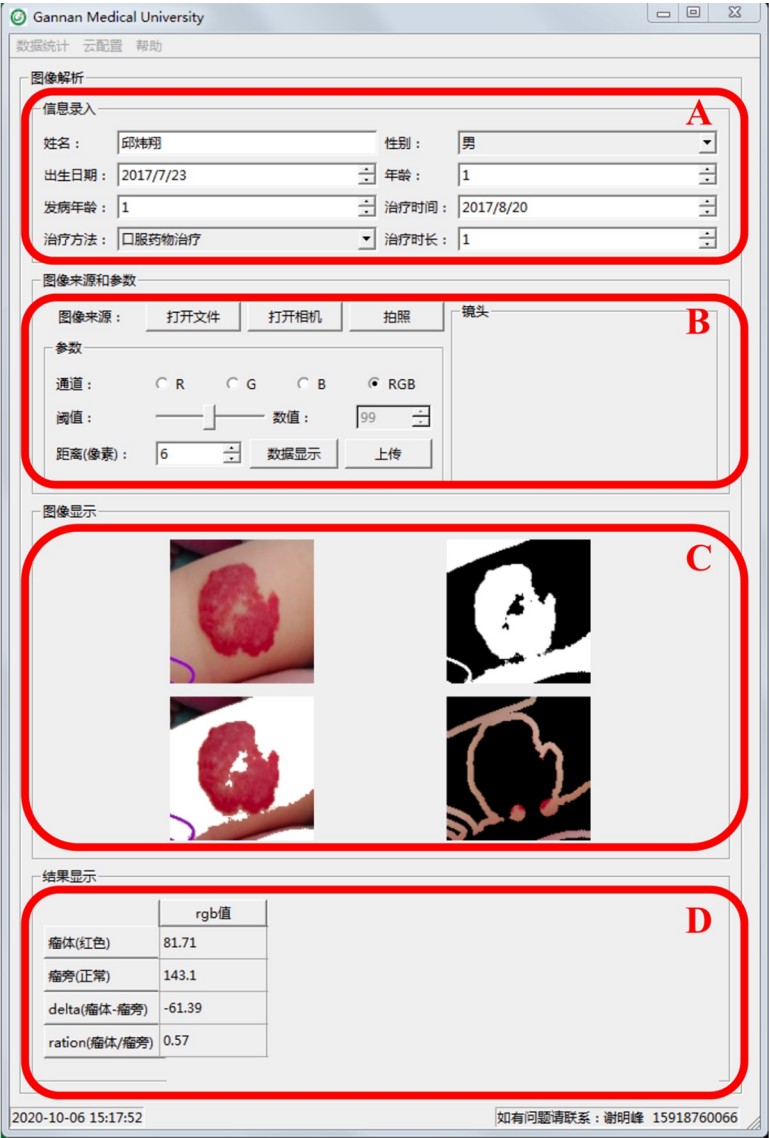

**Fig 3. Workflow used on the digital image analysis system.** (A) Patient information; (B) The RGB measurements; (C) Hemangioma images acquired for the patient; (D) The RGB result.

**Table 1. Clinical and demographic characteristics of the patients.**

| Clinical characteristic | No of patients / mean ± SD |
|---|---|
| Average age ± SD (months) | 3.48±1.96 |
| **Gender** | |
| Male | 68 |
| Female | 139 |
| Mean duration of treatment (months) | 3.52 ± 1.82 |
| **Lesion location** | |
| Head and neck | 61 |
| Limbs | 63 |
| Trunk | 75 |
| Perineum | 8 |

comprehensive tumor RGB values increased gradually, while the mean comprehensive RGB for the normal tissue around the tumor remained constant (Table 2). The difference in the RGB between the normal tissue and tumor decreased gradually after treatment, while the RGB coefficient increased, indicating a response to treatment (Table 2). This difference in the RGB ratio and the RGB difference were statistically significant for all-time points ($p<0.01$) after treatment, as shown in Figs 4 and 5.

## Correlation between the RGB ratio and other established methods used to grade IH

**RGB ratio versus the Achuaer method.** Table 3 summarizes the number of patients in our study in the five different grade groups proposed by Achuaer et al. before treatment and at various time points after treatment. As shown in Fig 6, the mean Achauer score before treatment was 1.14 ± 0.41. After one month and two months of treatment, the mean Achauer score increased to 2.17 ± 0.85 and 3.08 ± 0.92, respectively. The mean Achauer score reached 4.55 ± 0.64 at the end of the treatment. The difference in the mean Achauer score before and after each of the treatment time points was statistically significant ($p < 0.01$). The Spearman's correlation analysis revealed a strong positive correlation ($|\rho| = 0.670$; $p<0.05$) between the tumor RGB ratio and the Achauer score (Fig 7).

**RGB score versus the VAS scoring system.** Table 4 summarizes the number of patients within the nine VAS categories before treatment and at various time points after treatment. The mean VAS score before treatment was 7.88 ± 0.77 and decreased to 5.78 ± 0.74 after one month of treatment, 4.05 ± 0.77 after two months of treatment, and 2.80 ± 0.93 by the end of treatment (Fig 8). The difference between each time point was statistically significant ($p<0.01$). The Spearman correlation analysis revealed a statistically significant strong negative correlation between the VAS and RGB scores ($|\rho| = 0.756$; $p< 0.01$) (Fig 9).

**RGB score versus the HAS scoring method.** Table 5 summarizes the number of patients in each of the five HAS score categories before treatment and at various time points after

**Table 2. RGB mean, ratio, and difference of tumor and normal tissue around the tumor before treatment and at various time intervals after treatment.**

| RGB | Before treatment | One month after starting treatment | Two months after starting treatment | At the last treatment |
|---|---|---|---|---|
| **Tumor** | 90.14±29.35 | 103.40±31.77 | 112.68±31.98 | 131.42±33.93 |
| **Around tumor** | 146.93±39.57 | 143.66±35.92 | 141.07±34.22 | 144.93±36.40 |
| **Ratio** | 0.61±0.11 | 0.71±0.10 | 0.79±0.09 | 0.91±0.04 |
| **Difference** | -56.79±22.07 | -40.21±14.94 | -28.70±12.72 | -13.17±6.44 |

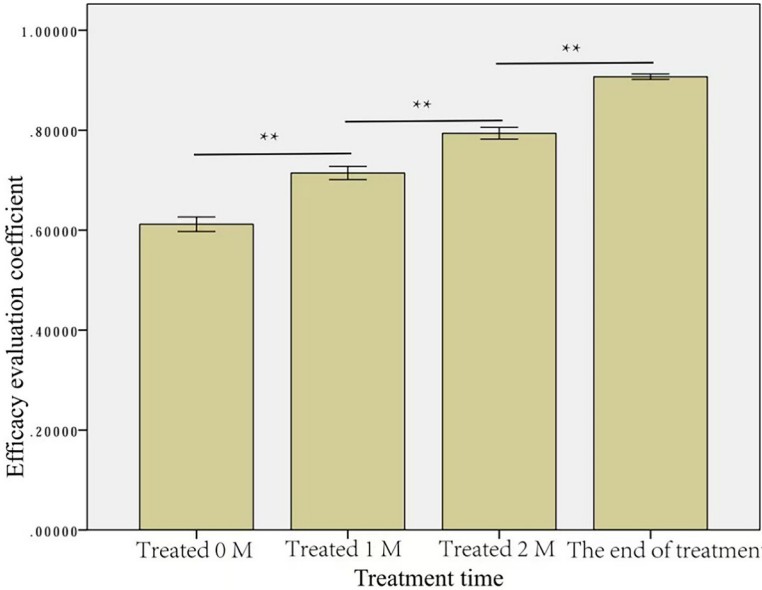

**Fig 4. RGB ratio between the tissue around the tumor before treatment and at various time points after treatment.** ** p<0.01.

treatment. The mean HAS score before treatment was 4.84 ± 0.45 and decreased to 3.79 ± 0.87 after one month of treatment, 2.89 ± 0.90 after two months of treatment, and reached 1.40 ± 0.55 by the end of the treatment. The difference between each time point was statistically significant (p < 0.01) (Fig 10). The Spearman correlation analysis revealed a statistically significant strong correlation between the RGB score and the HAS score (|ρ|-0.627, p<0.05) (Fig 11).

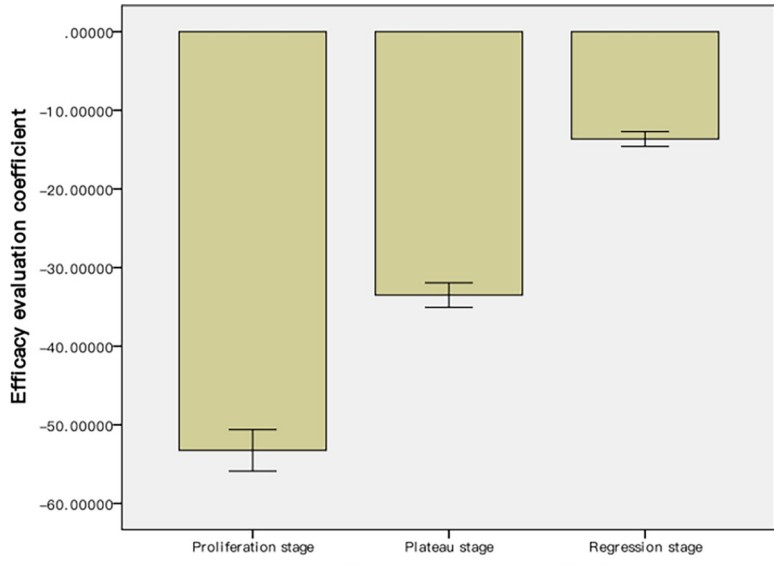

**Fig 5. RGB difference between the tissue around the tumor before treatment and at various time points after treatment.** ** p<0.01.

**Table 3. Number of patients within each Achuaer grade group.**

| | Before treatment | | One month after treatment | | Two months after treatment | | End of the treatment | |
|---|---|---|---|---|---|---|---|---|
| **Grade** | **n** | **%** | **n** | **%** | **n** | **%** | **n** | **%** |
| I | 184 | 88.9 | 37 | 17.9 | 7 | 3.4 | 0 | 0.0 |
| II | 18 | 8.7 | 117 | 56.5 | 44 | 21.3 | 0 | 0.0 |
| III | 5 | 2.4 | 38 | 18.4 | 95 | 45.9 | 16 | 7.7 |
| IV | 0 | 0.0 | 11 | 5.3 | 47 | 22.7 | 62 | 30.0 |
| V | 0 | 0.0 | 4 | 1.9 | 14 | 6.8 | 129 | 62.3 |
| **Mean±SD** | $1.14 \pm 0.41$ | | $2.17 \pm 0.85$ | | $3.08 \pm 0.92$ | | $4.55 \pm 0.64$ | |

*Grade I indicates severe hemangioma; grade 5 indicates normal skin.

**Correlation between Angiotensin II (Ang II) and RGB score.** Before treatment, the mean angiotensin II was $20.16 \pm 12.29$, and decreased to $16.06 \pm 4.30$ after one month of treatment, $12.30 \pm 4.12$ after two months of treatment, and reached $6.27 \pm 1.67$ by the end of treatment. The difference between each time point was statistically significant ($p < 0.01$) (Fig 12). The Pearson correlation analysis showed a statistically significant weak correlation between the Ang II levels and the RGB score ($|\rho| = 0.331$, $p < 0.05$) (Fig 13).

## Discussion

A change in the color of the tumor is an important indicator for treatment response in haemangioma patients. However, currently, there are no objective tools that could be used to quantify color changes in patients diagnosed with superficial IH. In this study, we proposed the first digital RGB scoring system to grade the severity of IH. The digital system evaluated treatment response by calculating the difference in the comprehensive RGB color between the

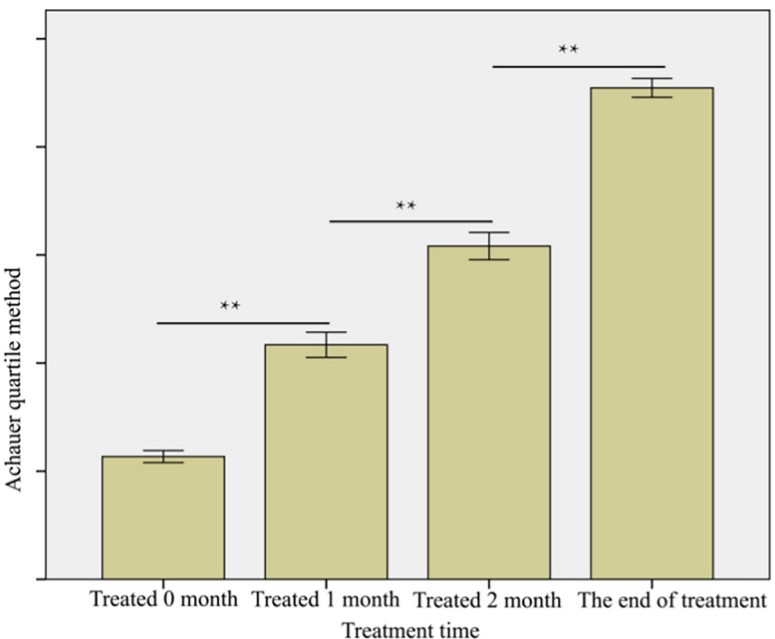

**Fig 6. Bar graph illustrating the number of hemangioma for each Achauer score (1 to 5) before and at various time points after treatment.** An Achauer score of 1 means severe hemangioma, and 5 means normal skin. ** $p < 0.01$.

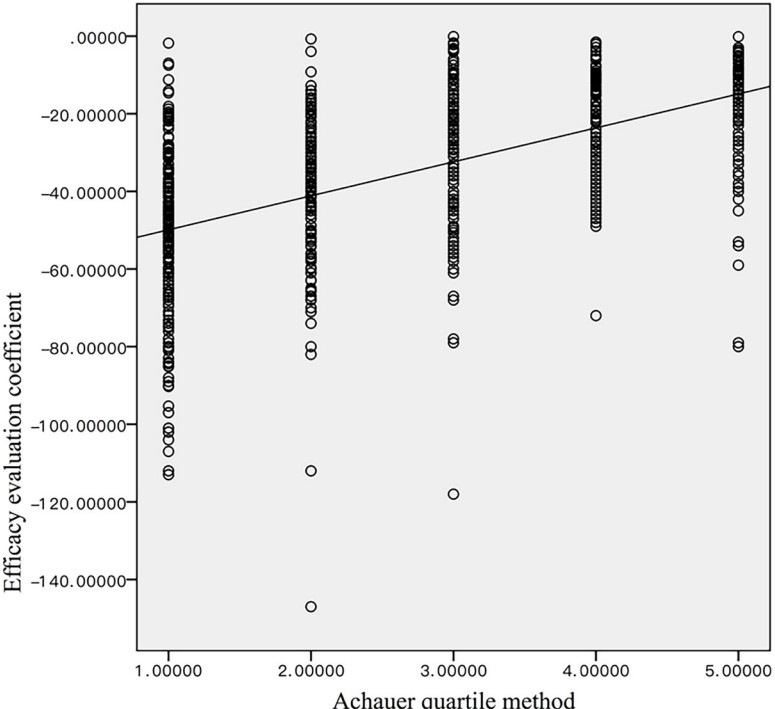

**Fig 7. The Spearman's correlation between the Achuar and RGB scores.**

tumor and surrounding normal tissue to account for individual variations in skin color. The system was then used to assess the treatment response in patients diagnosed with IH, and the results obtained were compared with other established visual and biochemical methods used to grade hemangiomas.

Our findings indicate that as the treatment progressed, the difference in the RGB score between the tumor and surrounding normal tissue decreased to reach almost zero by the end of the treatment. A value of zero means that there is no difference between the tumor bed and the surrounding normal tissue, indicating that treatment was successfully completed. These findings also show that clinicians could use the RGB score to assess tumor progression and

**Table 4. Number of patients within the VAS scores before and after treatment.**

|  | Before treatment | One month after treatment | Two months after treatment | End of the treatment |
|---|---|---|---|---|
| **VAS** | **n   %** | **n   %** | **n   %** | **n   %** |
| **9** | 44   21.3 | 0   0.0 | 0   0.0 | 0   0.0 |
| **8** | 101   48.8 | 0   0.0 | 0   0.0 | 0   0.0 |
| **7** | 56   27.1 | 34   16.4 | 0   0.0 | 0   0.0 |
| **6** | 6   2.9 | 97   46.9 | 1   0.5 | 0   0.0 |
| **5** | 0   0.0 | 72   34.8 | 63   30.4 | 0   0.0 |
| **4** | 0   0.0 | 4   1.9 | 88   42.5 | 58   28.0 |
| **3** | 0   0.0 | 0   0.0 | 55   26.6 | 65   31.4 |
| **2** | 0   0.0 | 0   0.0 | 0   0.0 | 69   33.3 |
| **1** | 0   0.0 | 0   0.0 | 0   0.0 | 15   7.2 |
| **Mean±SD** | 7.88 ± 0.77 | 5.78 ± 0.74 | 4.05 ± 0.77 | 2.80 ± 0.93 |

*VAS score of 1 indicates normal skin; VAS score of 9 indicates severe haemangioma.

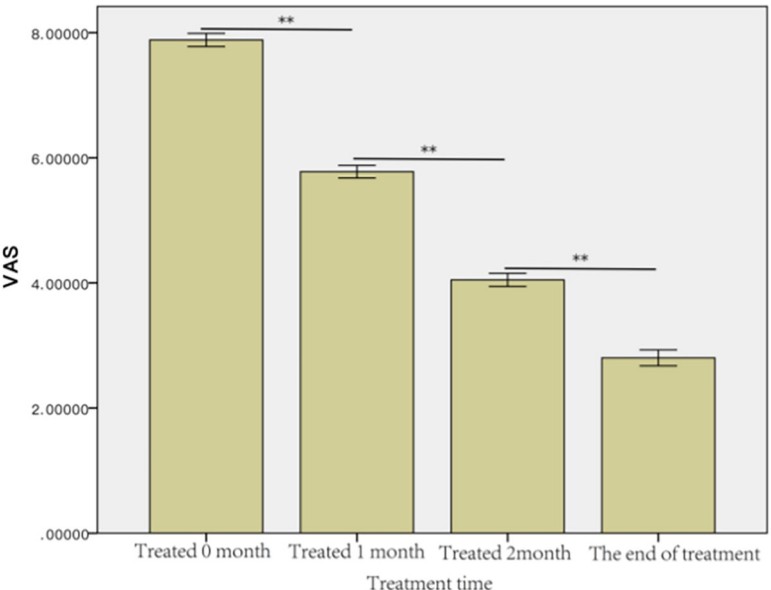

**Fig 8. VAS scores of the patients at each treatment time.** ** p<0.01.

evaluate treatment response at various points during the treatment. This information could be used by clinicians to optimize the treatment for the patient.

The RGB scoring system strongly correlated significantly with the Achauer, VAS, and HAS visual grading systems. However, the limitation of current visual grading systems is that they rely on visual assessments, which are highly subjective and cannot accurately and objectively

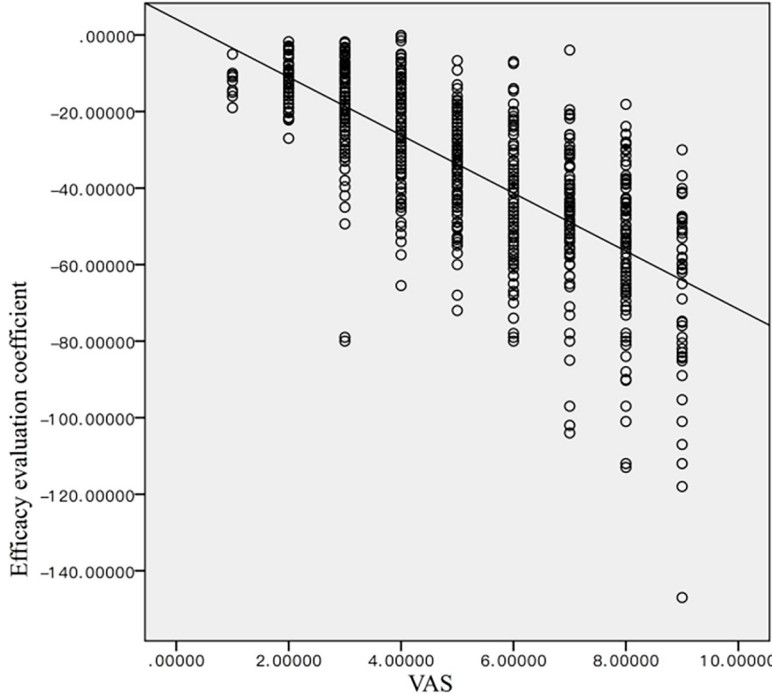

**Fig 9. The Spearman's correlation analysis between the VAS and RGB score.**

**Table 5. Number of patients in the five HAS score categories before treatment and at each time point after treatment.**

| | Before treatment | | One month after treatment | | Two months after treatment | | End of the treatment | |
|---|---|---|---|---|---|---|---|---|
| **HAS** | **n** | **%** | **n** | **%** | **n** | **%** | **n** | **%** |
| **5** | 181 | 87.4 | 37 | 17.9 | 7 | 3.4 | 0 | 0.0 |
| **4** | 19 | 9.2 | 109 | 52.7 | 47 | 22.7 | 0 | 0.0 |
| **3** | 7 | 3.4 | 47 | 22.7 | 100 | 48.3 | 6 | 2.9 |
| **2** | 0 | 0.0 | 11 | 5.3 | 47 | 22.7 | 71 | 34.3 |
| **1** | 0 | 0.0 | 3 | 1.4 | 14 | 6.8 | 130 | 62.8 |
| **Mean±SD** | 4.84 ± 0.45 | | 3.79 ± 0.87 | | 2.89 ± 0.90 | | 1.40 ± 0.55 | |

*HAS score of 1 indicates normal skin; HAS score of 5 indicates severe hemangioma.

describe the condition of the child. The superficial IH is often called "strawberry hemangioma" due to its typical shape, bright red color. We can assess the efficacy of treatment by changing the color of infantile hemangiomas [22]. The hemangioma tumor began to subside, its blood flow and vascular structure began to degenerate, the color gradually became lighter, and the color difference with the surrounding normal skin decreased and gradually became consistent [23]. At present, Achauer's [8] percentile quartile method and Zvulunov et al.'s [24] visual standard scoring method are mostly used to evaluate the color change of infantile hemangiomas. These methods are usually based on the observer's estimation, which is relatively subjective, and the results between different observers are inconvenient to compare. Therefore, in order to obtain accurate and objective data, it is necessary to quantify the color change of the tumor by this system.

Liu et al. [16] found that the renin-angiotensin-aldosterone system is closely related to the occurrence and development of IH. As a result, studies have shown that β-blockers can have an impact on the renin-angiotensin-aldosterone system. As a result, serum Ang-II level has been proposed as a biomarker for treatment response. However, our findings found a weak

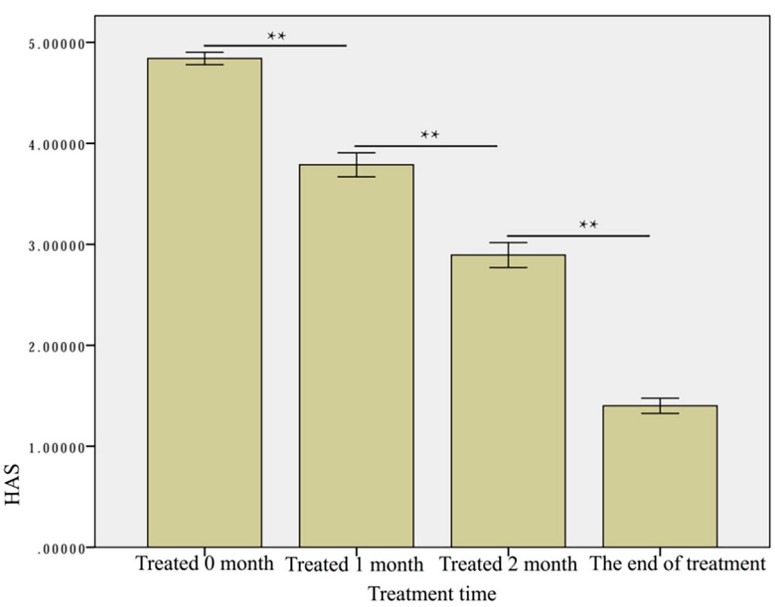

**Fig 10. The HAS scores of the patients at each treatment time.** ** $p < 0.01$.

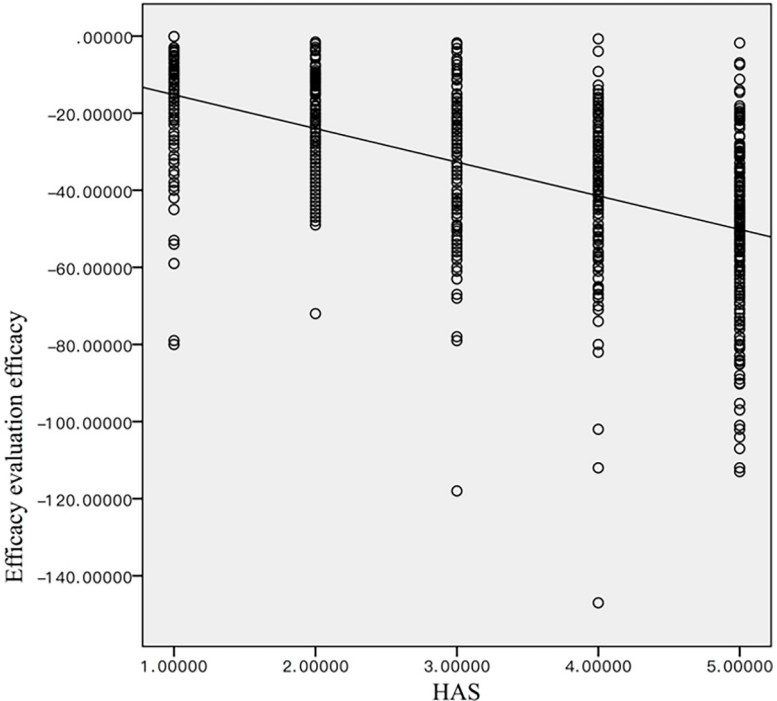

**Fig 11. The Spearman's correlation analysis between the HAS and RGB scores.**

correlation between the Ang-II levels and RGB score. A possible explanation could be that the Ang-II levels are significantly affected by external factors such as stress and drinking habits. Therefore, we believe that our RGB scoring system is a more accurate predictor of treatment response when compared with Ang-II.

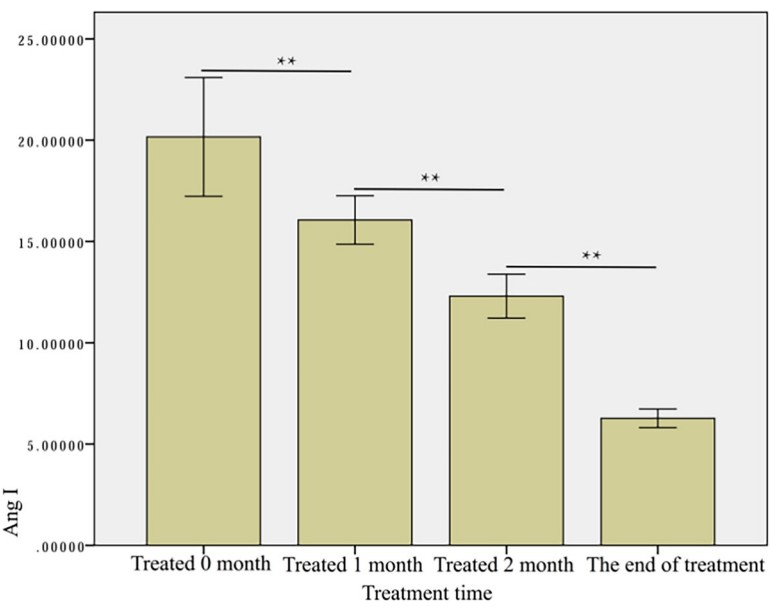

**Fig 12. Angiotensin II at each treatment time.** ** $p < 0.01$.

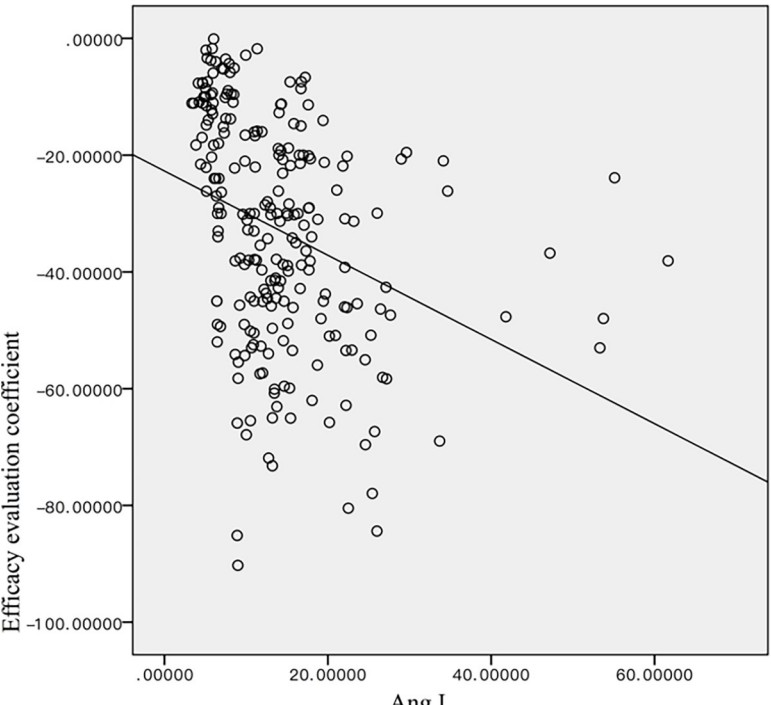

**Fig 13. The Pearson's correlation analysis between Ang II and the RGB score.**

## Conclusion

The RGB digital analysis system proposed in our study can objectively and accurately evaluate the color of the tumor in patients diagnosed with IH. However, more research is recommended to optimize the accuracy of this algorithm and its application in the assessment of other dermatological conditions such as radiation dermatitis.

## Supporting information

**S1 Dataset.**
(XLSX)

## Author Contributions

**Conceptualization:** Pingsheng Zhou, Qian Liu.

**Data curation:** Mingfeng Xie, Pingsheng Zhou, Yong Zeng.

**Formal analysis:** Mingfeng Xie, Haijin Huang, Wei Peng, Hui Xiao.

**Investigation:** Pingsheng Zhou, Haijin Liu, Haijin Huang, Hui Xiao.

**Methodology:** Qian Liu.

**Project administration:** Mingfeng Xie, Qian Liu.

**Resources:** Linshan Zeng, Feng Chen.

**Software:** Mingfeng Xie.

**Supervision:** Xianyun Xu, Qian Liu.

**Validation:** Jianping Liu.

**Visualization:** Jianping Liu.

**Writing – original draft:** Jianping Liu.

**Writing – review & editing:** Mingfeng Xie, Jianping Liu, Pingsheng Zhou, Qian Liu.

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
