## [Decision Letter · Decision Letter 0]

2 Feb 2023

PONE-D-22-29908Development of a digital imaging analysis system to evaluate the treatment response in superficial infantile hemangiomasPLOS ONE

Dear Dr. Liu,

Thank you for submitting your manuscript to PLOS ONE. After careful consideration, we feel that it has merit but does not fully meet PLOS ONE’s publication criteria as it currently stands. Therefore, we invite you to submit a revised version of the manuscript that addresses the points raised during the review process.

We look forward to receiving your revised manuscript.

Kind regards,

Fahmi Hussein Kakamad

Academic Editor

PLOS ONE

Journal Requirements:

Reviewers' comments:

Reviewer's Responses to Questions

**Comments to the Author**

1. Is the manuscript technically sound, and do the data support the conclusions?

Reviewer #1: Yes

Reviewer #2: Partly

2. Has the statistical analysis been performed appropriately and rigorously? 

Reviewer #1: Yes

Reviewer #2: Yes

3. Have the authors made all data underlying the findings in their manuscript fully available?

Reviewer #1: Yes

Reviewer #2: Yes

4. Is the manuscript presented in an intelligible fashion and written in standard English?

Reviewer #1: Yes

Reviewer #2: Yes

5. Review Comments to the Author

Reviewer #1: -A thorough English language revision is required.

-What is RGB appearing in the abstract, when an abbreviation appears for the first time, it should be defined.

-How the system helps in the clinical practice?! the authors should clearly address the benefit of this system

-Reference number 18,20,21 and 22 should be rewritten to become searchable reference

Reviewer #2: � The manuscript needs to be refined by a native speaker.

Introduction

1. Line 41. references are not listed in order of appearance in the text

2. Line 42. It is not appropriate to begin a sentence with an abbreviation. It must be corrected.

3. Line 66-96. The source from where you received this sentence should be cited.

4. The introduction part is typically very lengthy and needs to be brief.

Method

1. The method section has been well written and organized

2. Image acquisition:

Line 95. Please do not use abbreviation in the method section or at least write the complete word once before using abbreviation

3. Data acquisition: It is best to write inclusion criteria and exclusion criteria under different subheadings.

4. Ethical consideration: Ethics approval Please provide the ethics approval no.

Reference

The DOI should include references 4, 18, 20, 21, and 22.

Figure 2. A, B, C and D must be placed to the top left of each section of the figure, NOT overlapping the image

6. PLOS authors have the option to publish the peer review history of their article (what does this mean?). If published, this will include your full peer review and any attached files.

Reviewer #1: No

Reviewer #2: No

---

## [Author Response · Author response to Decision Letter 0]

8 Feb 2023

Dear Editor,

Thank you for reviewing our manuscript entitled "Digital imaging analysis system evaluating treatment response in superficial infantile hemangiomas". We appreciate your constructive comments and suggestions and have considered them in revising our work. Here is a point-by-point response to the reviewers' comments and concerns.

Comment 1:

The manuscript needs to be refined by a native speaker.

We appreciate your suggestion that the manuscript needs to be refined by a native speaker. We have asked a native English speaker to help us polish our article. We believe that this revision has greatly improved the readability and clarity of our manuscript.

Comment 2: Introduction:

A.Line 41. references are not listed in order of appearance in the text.

We agree with the reviewer's assessment and have revised the manuscript to ensure that all references are listed in the order of appearance within the text. 

B.Line 42. It is not appropriate to begin a sentence with an abbreviation. It must be corrected.

We appreciate your attention to detail and your suggestion regarding the use of abbreviations at the beginning of sentences. We have reviewed the text and made the necessary corrections to ensure that abbreviations are only used after they have been fully spelled out and defined.

C.Line 66-96. The source from where you received this sentence should be cited.

As suggested by the reviewer, we have added more references to support the statements in lines 66- 96. (references 8 and 18). 

D.The introduction part is typically very lengthy and needs to be brief.

We have carefully reviewed the section and removed any redundant or unnecessary information while still maintaining the essential elements required for a comprehensive understanding of the background and context of the study.

Comment 3: Method

A.The method section has been well-written and organized. Image acquisition. Line 95. Please do not use abbreviations in the method section, or at least write the complete word once before using abbreviations

We agree with the reviewer's assessment. We have reviewed the text to ensure that all abbreviations are defined at first occurrence within the text.

B.Data acquisition: It is best to write inclusion criteria and exclusion criteria under different subheadings.

We think this is an excellent suggestion and have now included the inclusion and exclusion criteria under a new subheading.

C.Ethical consideration: Ethics approval Please provide the ethics approval no.

As suggested by the reviewer, we have provided the ethics number in the manuscript.

Comment 4: References

The DOI should include references 4, 18, 20, 21, and 22.

Reference 4 was obtained from a book; therefore, the doi is unavailable. 

References 19 (previous 18), 21 (previous 20), and 22 (previous 21) are patents. We have now amended the format of these references to align with the PLOS ONE guidelines.

Reference 23 (previous 22) was obtained from a journal. We have now included the DOI for this reference.

Comment 5: Figures 

Figures 2. A, B, C, and D must be placed to the top left of each section of the figure, NOT overlapping the image

We have now reformatted figure 2 so that the labels do not obscure the images.

Once again, we would like to thank you for your time and expertise, and we hope that the revised manuscript meets your expectations.

Sincerely,

Qian Liu

The First Affiliated Hospital of GanNan Medical University

Jiangxi University of Traditional Chinese Medicine

Ganzhou, Jiangxi, China

liuqianjxzyy@163.com

---

## [Editor Report · Decision Letter 1]

13 Feb 2023

Development of a digital imaging analysis system to evaluate the treatment response in superficial infantile hemangiomas

PONE-D-22-29908R1

Dear Dr. Liu,

We’re pleased to inform you that your manuscript has been judged scientifically suitable for publication and will be formally accepted for publication once it meets all outstanding technical requirements.

Kind regards,

Fahmi Hussein Kakamad

Academic Editor

PLOS ONE
---

## [Editor Report · Acceptance letter]

12 Mar 2023

PONE-D-22-29908R1 

Development of a digital imaging analysis system to evaluate the treatment response in superficial infantile hemangiomas 

Dear Dr. Liu:

I'm pleased to inform you that your manuscript has been deemed suitable for publication in PLOS ONE. Congratulations! Your manuscript is now with our production department. 

Kind regards, 

on behalf of

Dr. Fahmi Hussein Kakamad 

Academic Editor

PLOS ONE